# Impact of host factors and invasive meningococci on bacterial adhesion, proliferation, primary nasal epithelial barrier function, and immune response

Daan W. Arends,[1,2] Debbie van Rooijen,[1] Esther van Woudenbergh,[1,2] Janine Wolf,[1] Milou Ohm,[1] Marien I. de Jonge,[2] Gerco den Hartog[1,2]

**ABSTRACT** *Neisseria meningitidis* colonizes the human upper airway mucosa, which can progress into invasive meningococcal disease (IMD) upon breaching the epithelial barrier. Some serogroups and lineages are associated with IMD, whereas others rarely infect patients. Although multiple virulence factors have been described, it is unclear what makes some lineages hyperinvasive. Here, we examined meningococcal infection of air-to-liquid interface (ALI)-differentiated primary epithelial cells to assess host- and strain-dependent differences in colonization that could explain hyperinvasiveness. Nasal epithelial cells from seven donors were infected with meningococcal reference strains and hyperinvasive clonal complex 11 (cc11) strains, including serogroup C, W, and an unencapsulated capsule null locus strain. We assessed transepithelial electrical resistance (TEER), bacterial binding and growth, and the epithelial cytokine response. Most donor-strain combinations induced a TEER drop, with higher bacterial loads generally associated with lower TEER, indicative of a more permeable epithelial barrier. Cc11 strains induced slightly lower TEER levels than other strains. Bacterial binding and growth were highly donor-strain dependent. The unencapsulated strain exhibited the highest binding, and MenW cc11 showed higher binding than the MenW reference strain, but the MenC strains showed no difference. Despite strain and donor variability, meningococcal infection universally induced cytokines CCL20, CXCL1, CXCL8, CXCL10, and IL-18, with significantly higher CCL20 levels 24 h post-MenW cc11 infection. Principal component analysis showed cytokine profiles were predominantly influenced by the epithelial cell donor rather than the meningococcal strain. In conclusion, our experimental model shows that the outcome of meningococcal infection of the epithelium is highly dependent on specific donor-strain interactions.

**IMPORTANCE** Invasive meningococcal disease (IMD) outbreaks remain a significant disease burden, often caused by *Neisseria meningitidis* strains belonging to "hyperinvasive" lineages, such as clonal complex (cc) 11. To determine what factors contribute to the invasiveness of a meningococcal strain, we examined the initial stage of becoming invasive, the interaction between upper airway epithelium and the bacteria. To assess both the role of the infecting meningococcal strain and host epithelium composition, we used multiple meningococcal strains and nasal primary epithelial cells from several healthy individuals. Our observation of specific infection dynamics for each cell donor and meningococcal strain combination highlights the importance of both host and bacterial factors—and especially their interaction—in the determination of the outcome of meningococcal infection. Understanding these host-strain-specific effects could help to identify individuals at risk of IMD and meningococcal strains with future outbreak potential.

**Peer Reviewers** Mathieu Coureuil, Institut Necker-Enfants Malades, Paris, France; Christopher D. Bayliss, University of Leicester, Leicester, United Kingdom

Address correspondence to Daan W. Arends, daan.arends@rivm.nl, or Gerco den Hartog, gerco.den.hartog@rivm.nl.

The authors declare no conflict of interest.

See the funding table on p. 15.

**KEYWORDS** *Neisseria meningitidis*, host-pathogen interactions, primary epithelial cells, respiratory pathogens, gram-negative bacteria, meningococcus

*Neisseria meningitidis* is a human-specific gram-negative diplococcus that colonizes the mucosa of the upper airways. Colonization mainly occurs asymptomatically where meningococci stay luminal and do not cause any noticeable inflammation. However, colonization can progress into invasive meningococcal disease (IMD) after epithelial barrier breach and access to the bloodstream. Meningococcal invasion can result in septicemia and meningitis and is associated with high mortality rates (5–10%), especially in people ≥60 years of age (15–20%) (1, 2). IMD can also lead to sequelae such as amputations, scarring, hearing loss, anxiety, and psychological developmental difficulties (3). IMD is most prevalent in young children and people ≥65 years of age, suggesting that host immune status is an important determinant. However, the prevalence of IMD is also relatively high in adolescents, resulting in high mortality rates (8–10% in 15-24 years [2]), and it is still not entirely clear why (4).

Since 2000, two major outbreaks of IMD have occurred in the Netherlands: in 2000–2002 caused by serogroup C (5) and in 2017–2018 by serogroup W (6) meningococci. In both periods, most of the isolates belonged to clonal complex 11 (cc11), a meningococcal lineage that has caused multiple IMD outbreaks throughout the world since the 1960s, garnering it a hypervirulent reputation. In contrast to most other lineages that are associated with one type of polysaccharide capsule, due to multiple capsule switching events, cc11 isolates have been found expressing either serogroup B, C, W, or Y capsule (7, 8). The MenW cc11 variant that caused the 2017–2018 outbreak emerged in the United Kingdom in 2013 and has caused disease across Europe (9). The reasons behind the increased capacity of cc11 to cause invasive disease are not completely clear.

A prerequisite of invasive disease is successful colonization of the airway mucosa, which the meningococcus achieves by expressing adhesion factors such as type IV pili and adhesins Opa, Opc, and NadA, which bind receptors on the apical side of epithelial cells (10). In addition, the release of extracellular DNA by living bacteria or through autolysis is an important factor in biofilm formation (11). Phase variation alters the expression of surface-exposed proteins, allowing the meningococcus to evade host immune responses (12, 13). Meningococci also express proteins that interfere with specific host defense mechanisms at the mucosa, such as IgA protease IgA1P (14, 15) and factor H binding protein (16), inhibiting neutralization and complement deposition.

To become invasive, meningococci must breach the epithelial barrier consisting of: (i) a layer of mucus secreted by submucosal glands and specialized secretory cells within the epithelium, which facilitates mucociliary clearance of bacteria and which contains antimicrobial peptides (AMPs), making it an inhospitable environment for some bacterial species; and (ii) a tightly packed layer of cells that through tight and adherence junctions, form a mechanical barrier for micro-organisms (17, 18). After breaching the epithelial barrier, the progression of invasive disease is determined by meningococcal growth and inadequate host immune responses. Here, meningococcal encapsulation, meningococcal load, the functionality of the human complement system, and acquired antibodies are essential (19, 20). Besides forming a barrier, the epithelium is central to the initial immune response against potential pathogens. It is able to sense the presence of bacteria, and its response can range from tolerance of its presence to induction of an active immune response against it, by secretion of multiple cytokines (17, 18).

Currently, it is unknown to what extent specific intrinsic host differences in the epithelial cell response, as well as meningococcal strain-specific responses of the epithelium, determine whether meningococcal colonization progresses into invasion. In this study, we examined the role of the epithelial cells and meningococcal strains in the progression of colonization and the initial epithelial response. We infected primary nasal epithelial cells from seven different donors with multiple invasive meningococcal isolates belonging to clonal complex cc11. We assessed the epithelial barrier integrity

during infection, the bacterial load on the epithelial layers, and the cytokine response induced upon infection.

## RESULTS

### Epithelial barrier integrity

Epithelial cells of seven donors were collected as described in Materials and Methods (Table 1). Primary epithelial cells were differentiated on an air-liquid interface (ALI) for 4–6 weeks. Fluorescence microscopy analysis of these layers is described by van Woudenbergh et al. (21). In short, epithelial layers were of similar cross-sectional thickness, with the exception of D5, which was markedly thinner. Less Tubulin IV signal (cilia) could be observed in 6-week-old epithelium from D3 and D5. D1, D2, and D7 epithelium contained slightly more SCGB1-A1-positive cells (club cells). We first examined the mechanical barrier function of these epithelial layers in uninfected conditions by measuring the transepithelial electrical resistance (TEER). We observed significant differences between epithelial layers originating from different primary cell donors (Fig. 1A), with over fourfold difference between some donors.

To assess the effect of meningococcal infection on epithelial barrier integrity, we infected the epithelial layers with lineage cc11 MenW and MenC strains, with two laboratory MenW and MenC strains routinely used for serum bactericidal assays, and with an unencapsulated (Cnl) strain (Table 2). After 24 h, TEER was lower in almost all infected inserts compared to prior to infection, suggesting impaired epithelial barrier integrity, except for wells containing epithelial layers derived from donor 3 (Fig. 1B). Additionally, layers derived from donor 5 showed a similar decrease in both uninfected and infected conditions. TEER development, expressed in median TEER fold change after 24 h of infection, differed significantly when the different meningococci were compared (Fig. 1C). Both cc11 strains were lower in general than the other tested strains, but a pairwise significant difference was only reached for the MenW strain. When meningococcal strains were stratified by either serogroup or sequence type, no significant difference in TEER fold change was observed when uninfected results were excluded (data not shown). Median fold change in TEER during infection also differed between epithelial cell donors (Fig. 1D), with significant pairwise differences between multiple donors. For infection of epithelial layers of donors D5–D7, we included six additional strains, which affected TEER development similarly (Fig. S1). Based on this, we conclude that the observed TEER change is more dependent on epithelial cell origin than on the meningococcal strain.

### Meningococcal binding and bacterial load during infection

Differences in the ability of meningococcal strains to colonize are influenced by their capacity to bind to and/or grow on the epithelial layer. To assess the differences in binding capacity of the meningococci to the epithelial cells, we measured the number of CFUs that were bound to the epithelial cell layer 2 h after inoculation. The percentage of the inoculum that was bound differed significantly between strains (Fig. 2A), with the highest levels of binding found for the unencapsulated Cnl strain. The MenW cc11 strain differed significantly from the MenW reference strain. However, no difference was

**TABLE 1** Epithelial cell donor characteristics

| Donor | Sex | Age | Nasal origin |
| --- | --- | --- | --- |
| D1 | Female | 17 | Mucoid tissue nose septum |
| D2 | Female | 5 | Nasal epithelium of nose cavity |
| D3 | Female | 28 | Concha inferior |
| D4 | Female | 61 | Nasal epithelium of nose cavity |
| D5 | Male | 17 | Concha inferior |
| D6 | Female | 38 | Concha inferior |
| D7 | Female | 35 | Bulla thmoidalis |

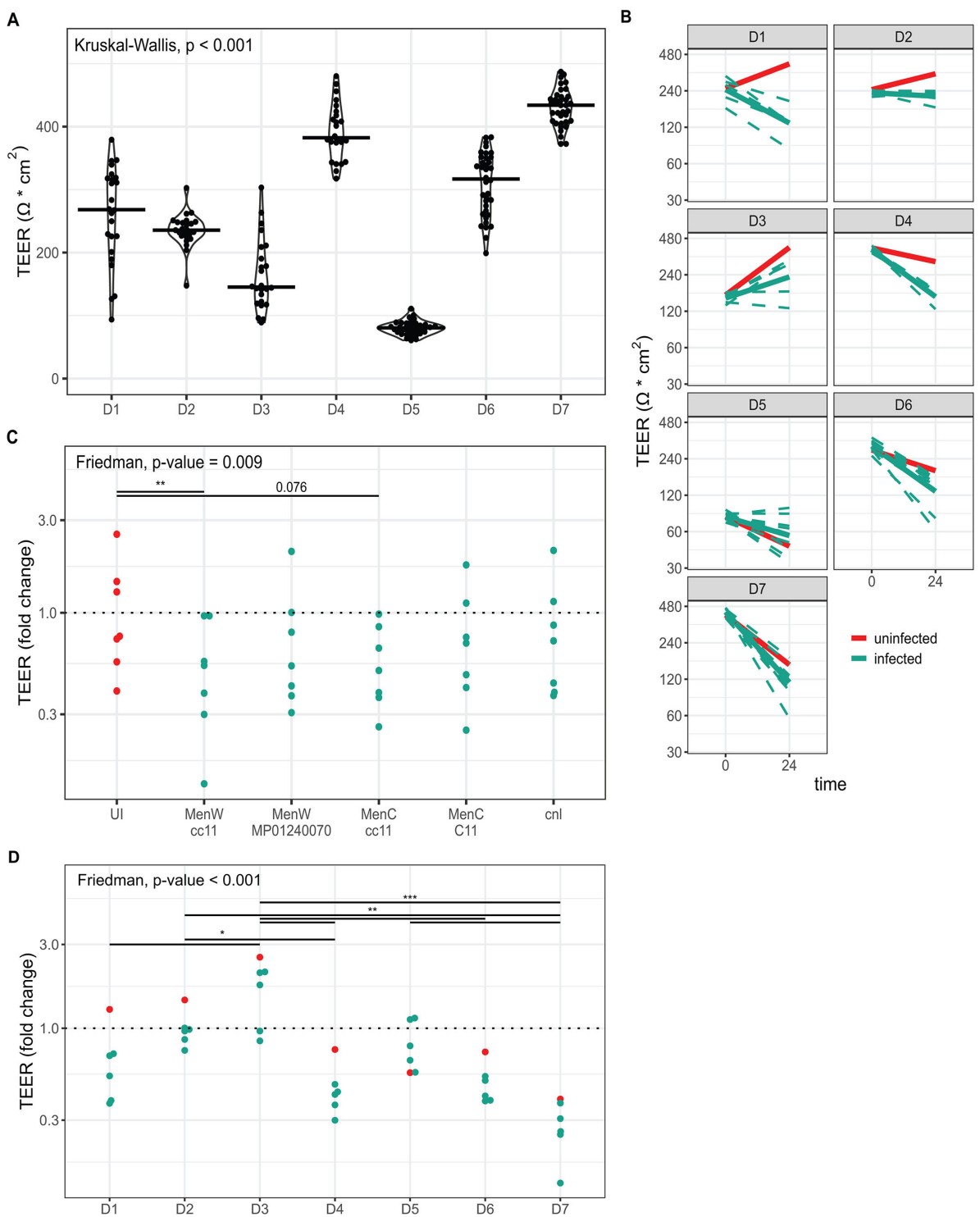

FIG 1 Transepithelial electrical resistance (TEER) development during meningococcal infection. (A) Baseline TEER with median of $n = 24$ (D1-D4) or $n = 36$ (D5-D7) uninfected epithelial at t = 0. (B) TEER of epithelial layers over time in uninfected (red) and infected (blue) conditions. Dotted lines represent the mean of triplicate measurements (D1–D4) or single measurements (D5–D7) of individual strains. Solid blue lines represent the mean of all infected conditions. TEER fold change after 24 h of infection stratified by infecting meningococcal strain (C) or epithelial cell donor (D). t = 24 dots represent mean of triplicate (D1–D4) or single (D5–D7) measurements. Performed statistical tests: Kruskal-Wallis test (A), permutation-based Friedman test (C, D), Dunn's post hoc test with Benjamini-Hochberg multiple comparison correction (C, D). Dunn's test P values are grouped by level of significance, where *, **, and *** indicate P values of less than 0.05, 0.01, and 0.001, respectively. UI, uninfected.

**TABLE 2** Meningococcal strains

| # | Strain | Strain ID | Serogroup | Clonal complex | ST | Year of isolation |
|---|--------|-----------|-----------|----------------|-----|-------------------|
| 1 | MenW cc11 | 46966 | W | 11 | 11 | 2012 |
| 2 | MenW MP01240070 | MP01240070 | W | 22 | 184 | – |
| 3 | MenC cc11 | – | C | 11 | 11 | 2002 |
| 4 | MenC C11 | C11 | C | – | 345 | – |
| 5 | Cnl | – | – | 1,136 | 1,136 | 2018 |
| 6 | MenW cc11_2 | 61465 | W | 11 | 11 | 2018 |
| 7 | MenW cc11_3 | 45363 | W | 11 | 11 | 2016 |
| 8 | MenW cc22_1 | 44661 | W | 22 | 3,422 | 2015 |
| 9 | MenW cc22_2 | 55103 | W | 22 | 3,422 | 2017 |
| 10 | MenC cc11_2 | – | C | 11 | 11 | 2017 |
| 11 | MenC cc11_3 | – | C | 11 | 11 | 2001 |

observed between the two MenC strains. Binding also differed significantly between epithelial cell donors (Fig. 2C), where epithelial layers derived from D3 showed notably lower levels of binding. Binding of a larger selection of strains, including other sequence types, was examined on RPMI-2650 (nasal septum carcinoma) and Calu-3 (lung adenocarcinoma) cells because these are easier and more cost-effective to culture. Further examination of six strains that had consistent high or low binding on RPMI-2650 and Calu-3 cells showed that these data did not reflect binding on physiologically more representative primary epithelial cells (D5–D7) (Fig. S2) and are therefore not further examined.

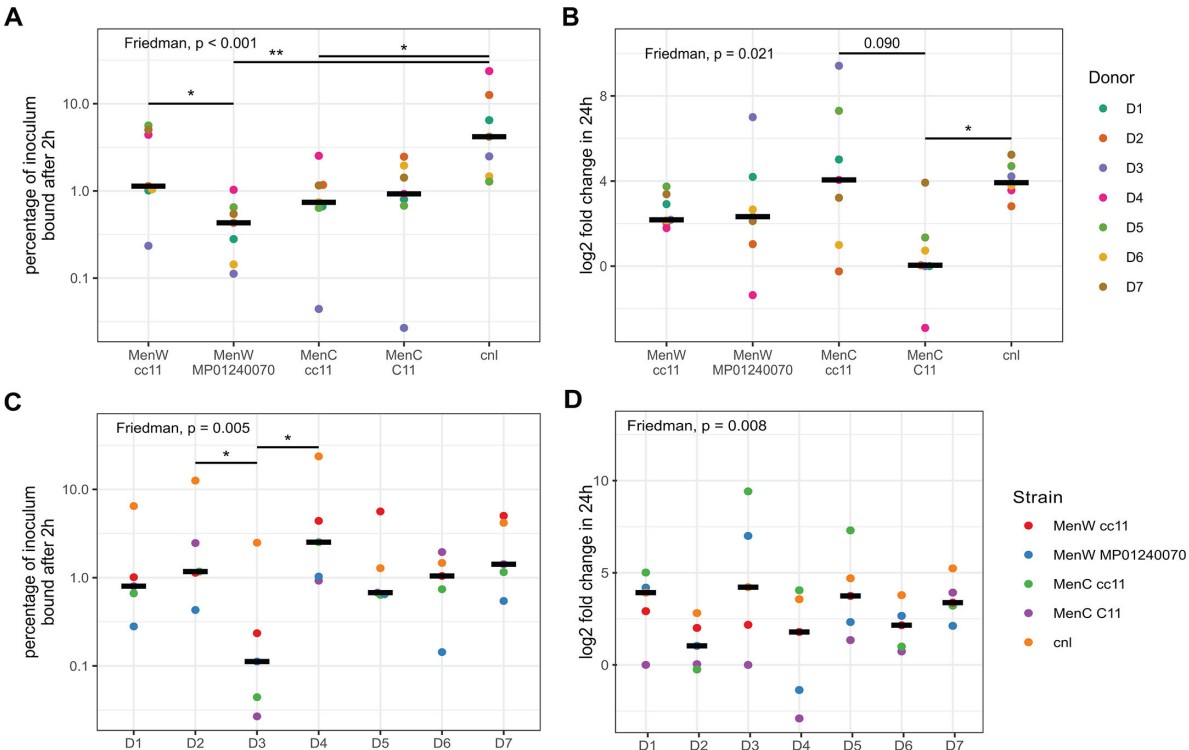

**FIG 2** Meningococcal attachment to and growth on the epithelial cell layers. Percentage of the meningococcal inoculum ($1*10^5$ CFU [MOI ~ 1]) bound to the epithelial cell, based on colony forming units (CFUs) stratified by infecting strain (A) or epithelial cell donor (C). Meningococcal growth in 24 h of infection (fold change compared to $t = 0$) stratified by infecting strain (B) or epithelial cell donor (D). Dots represent the individual donor (A, B) or each unique strain (C, D). Line indicates median percentage or fold change. Performed statistical tests: permutation-based Friedman test.

Next, we assessed whether bacterial growth during infection was different between strains and donors by calculating the fold change in CFU from $t = 0$ h to $t = 24$ h. Again, significant strain and donor differences were found (Fig. 2B and D, respectively). The MenC cc11 strain displayed the highest levels of growth, approaching significance compared to the MenC C11 strain ($P = 0.090$). However, it differed substantially between epithelial cell donors, as can be observed in its large CFU fold change spread. Some strains displayed little difference between donors, suggesting that for these strains, the epithelial layer phenotype has little effect on bacterial growth. Similarly, the epithelial cell donor effect on growth showed different levels of variation between meningococcal isolates, suggesting that specific donor-strain interactions are responsible for the measurement of extreme fold changes.

Besides relative changes in growth expressed as fold change, the absolute bacterial load is also of importance. Log-transformed linear regression of all infected samples showed a small negative association between CFU and TEER 24 h post-infection ($R^2 = 0.12$, $P = 0.01$; Fig. 3A). When specific meningococcal strains and epithelial cell donors are examined, such a correlation is less obvious. To examine the interplay of bacterial binding, bacterial load at 24 h and epithelial barrier function, these data were stratified for both factors in Fig. 3B. Different outcomes of epithelial infection can be observed. Notable observations are the high levels of Cnl binding and bacterial load at 24 h, but with relatively small effects on TEER change. Also remarkable are the high bacterial loads on donor D3 cells despite the low binding, and the often relatively large effects on TEER for both cc11 strains, especially MenW cc11. However, no universal effect can be observed for a specific meningococcal strain or epithelial cell donor, highlighting the influence of specific donor-strain combinations on TEER and bacterial loads.

## Epithelial cytokine response

Possibly, these different interactions also result in differences in epithelial immune signaling. To examine the epithelial cytokine response upon infection, medium from the basolateral side of the epithelium was collected 6, 24, and 48 h after infection. We determined the concentration of 24 cytokines (see Materials and Methods). Of these, CCL20, CXCL10, CXCL1, CXCL8, and IL-18 showed higher levels compared to uninfected epithelium 24 h after infection (Fig. 4). The MenW cc11 strain induced significantly higher levels of CCL20 than all other encapsulated strains. After 6 h, no significant differences were observed for any cytokine. As the 48 h time point consisted of a single measurement, no statistical tests were performed, but overall concentrations followed the trend that was observed at 24 h. At 48 h, MenW cc11 strain-induced CCL20 levels returned to levels similar to those of other strains. Besides CCL20, no significant differences between strains, serogroups, or lineage were found for any of the cytokines, suggesting a mostly universal cytokine response upon meningococcal infection. An increase in basolateral cytokines was also observed in uninfected wells, likely the result of baseline cytokine secretions that accumulate in the enclosed space of the well. We did not observe a clear induction of important proinflammatory cytokines IL-1β, IL-6, and TNFα. IL-1β was significantly higher 24 h post-infection, but the difference was small, and 48 h post-infection concentrations were no different from uninfected conditions. IL-6 concentration was the same for all conditions at 24 h post-infection, but at 48 h, concentrations were higher in infected inserts. TNFα concentrations were close to the detection limit and similar for all conditions. Principal component analysis (PCA) of a subset of 14 cytokines that could be measured 24 h post-infection in the samples of all donors revealed a very significant role (permutational multivariate analysis of variance [(PERMANOVA)]: $R^2 = 0.51$, $P < 0.001$; Fig. 5A) for the epithelial cell donor on the overall cytokine profile. The strain effect was small and nonsignificant ($R^2 = 0.08$; Fig. 5B). Examination of squared cosine of the PCs showed that usually a set of specific cytokines was responsible for the variation explained by each PC (Fig. 5C). Notable significant differences between donors in cytokine concentration could be observed for CCL2, CCL20, CXCL1, CXCL5, CXCL8, and IL-6 (Fig. S3).

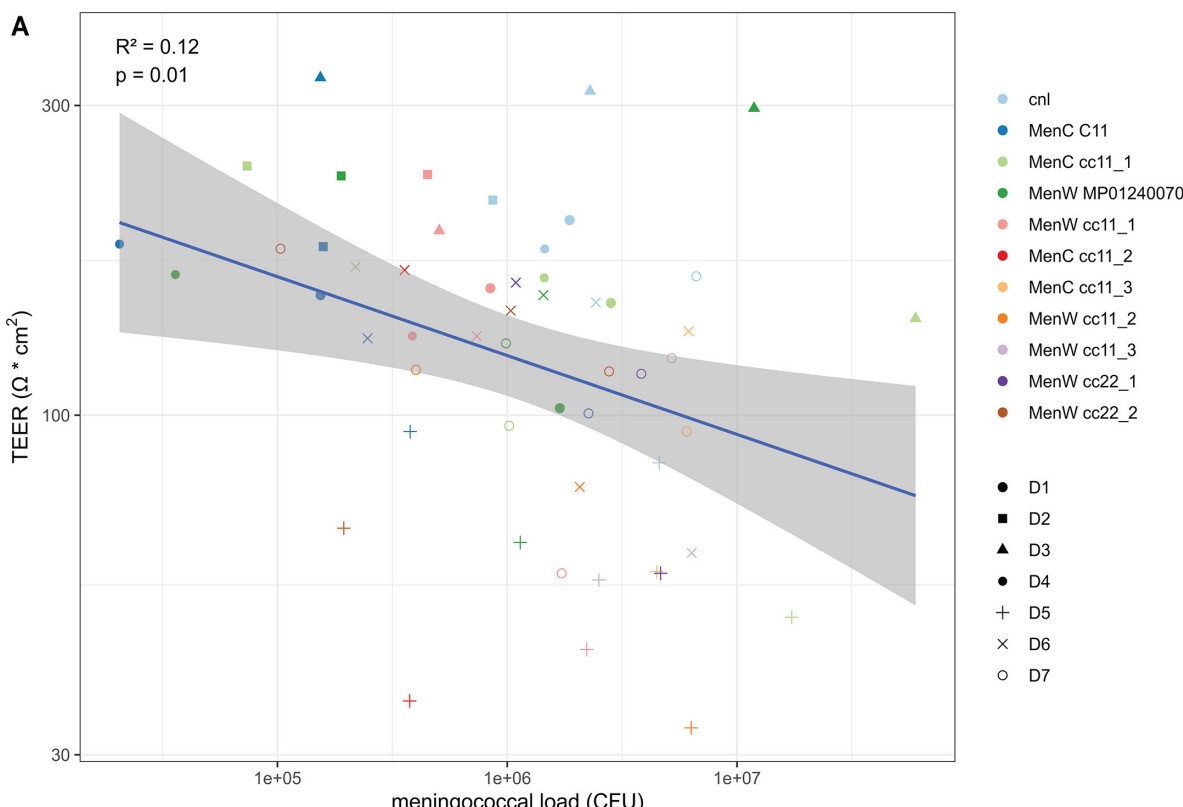

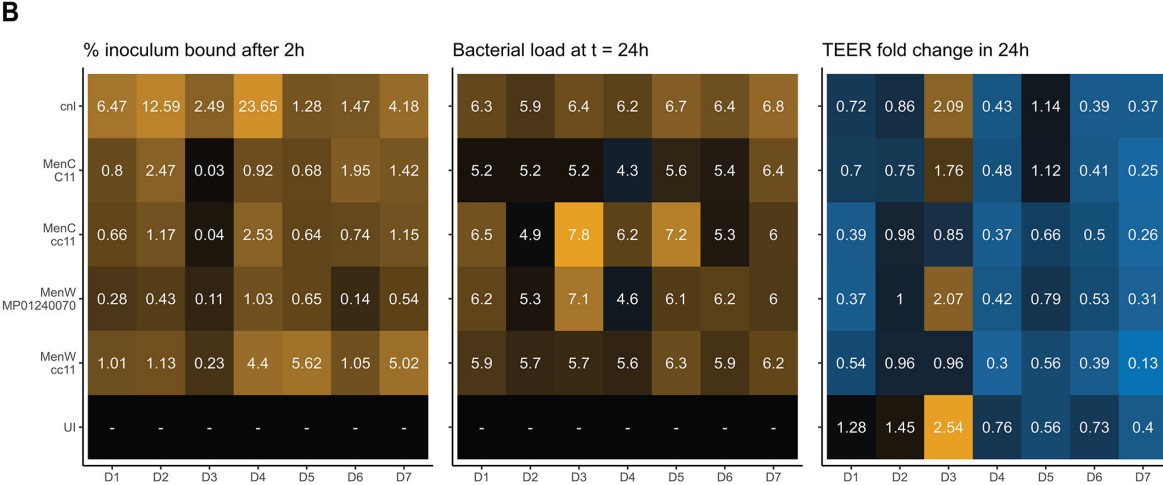

**FIG 3** Association between bacterial load (CFU) and epithelial barrier integrity (TEER). (A) Log-transformed linear regression of the effect of CFU on TEER 24 h post-infection, showing epithelial cell donor (dot shape) and meningococcal strain (color). (B) Heatmaps showing mean percentage of inoculum bound (left; color intensity increases with percentage), bacterial load expressed as $\log_{10}$(CFU) (middle; color indicates level below (blue) or above (yellow) initial inoculum CFU), and TEER fold change (right; color indicates TEER decrease (blue) or increase (yellow) 24 h post-infection), for each specific strain-donor pairing. UI, uninfected.

## DISCUSSION

How colonization and infection of the upper respiratory epithelium by meningococcal isolates may contribute to the invasiveness of specific strains is incompletely understood. The outcome of meningococcal colonization depends on several factors: the ability of the meningococcus to evade host antimicrobial responses and establish a niche by securing nutrients and resisting mucociliary clearance; the ability of the host to induce

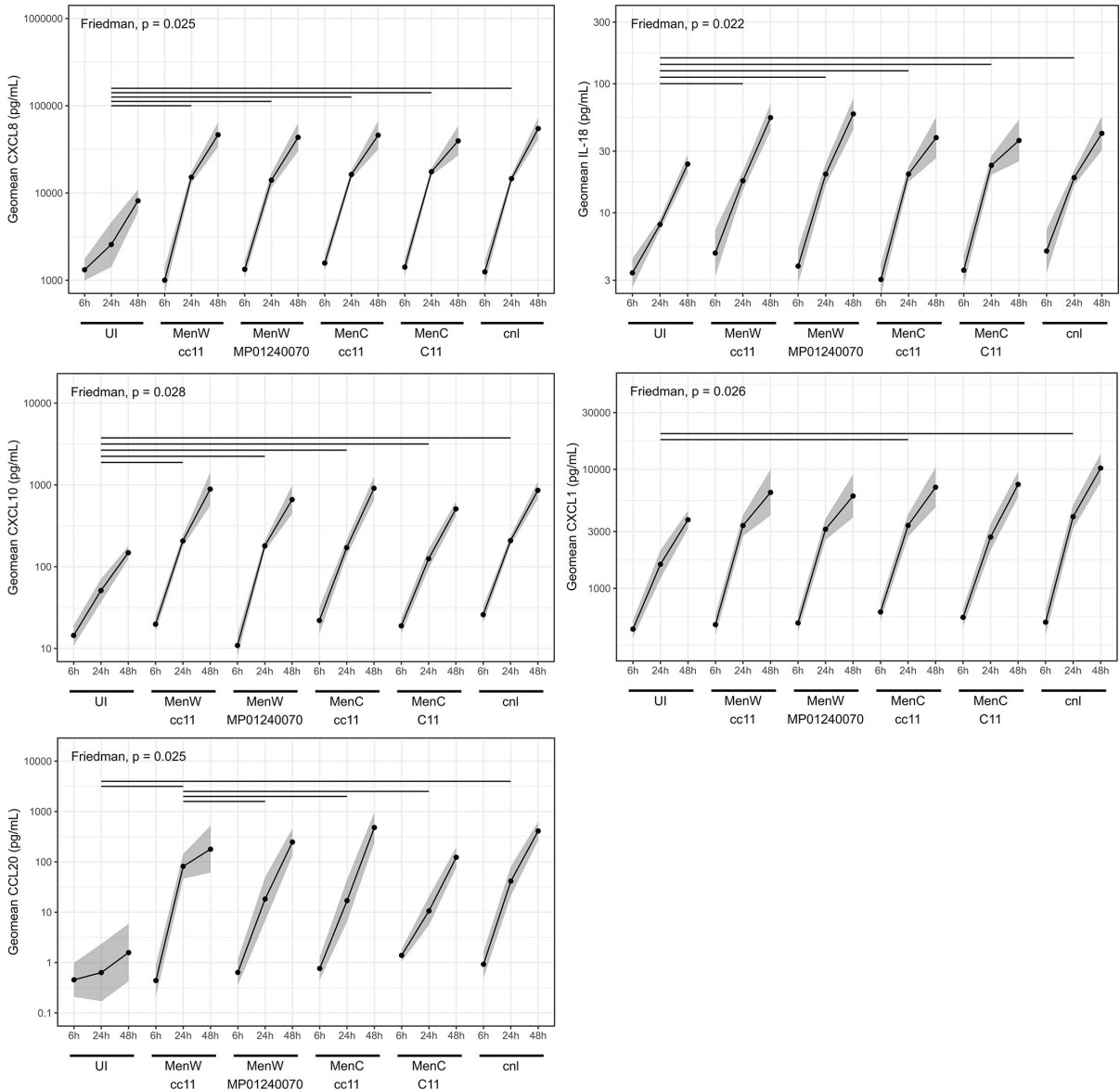

**FIG 4** Universal epithelial cytokine response upon meningococcal infection over time. For uninfected (UI) cells and each individual strain, cytokine concentrations at 6, 24, and 48 h post-infection are shown. Lines represent the mean geomean concentration across all donors with 95% CI. Statistically significant differences were observed 24 h post-infection. Performed statistical analysis: per strain, the mean concentration of individual donors was calculated at 24 h post-infection. These were then compared using a permutation Friedman test stratified by donor and post hoc testing with a Dunn's test using Benjamini-Hochberg for multiple analysis correction. Lines indicate significantly different pairings (at least $P < 0.05$).

an appropriate response, the maintenance of the epithelial barrier, and the elimination of meningococci that breach the barrier. Here, we have shown that during infection of primary epithelial cell layers with an ALI, the interaction between host epithelium and meningococcal isolate seems to be the most important factor in infection progression.

We observed substantial differences in epithelial layer baseline TEER and TEER fold change during infection using an MOI of 1. The effect of meningococcal infection on TEER varied depending on the donor. We observed slightly lower TEERs for cc11 strains after 24 h of infection. Across all donors and strains, we observed a negative correlation for CFU and TEER levels, indicating that higher bacterial loads might impair epithelial barrier integrity, or that increased barrier leakage enables greater bacterial growth. The observed decrease of TEER during infection may be caused by loss of tight junctions

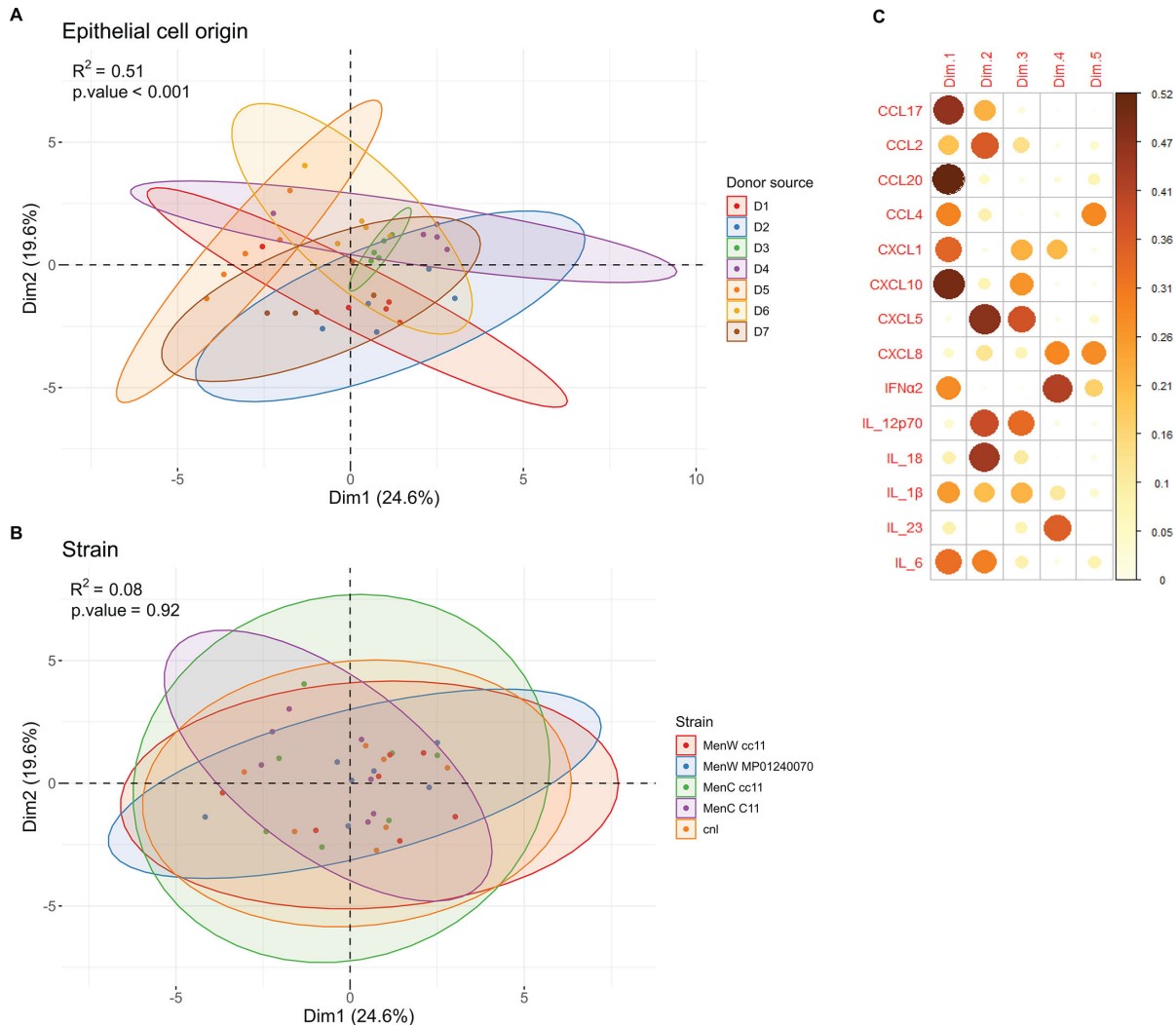

**FIG 5** Principal component analysis of cytokine response profile after 24 h of meningococcal infection, excluding uninfected samples and cytokines that were mostly undetected. (A) Epithelial cell donor was responsible for most of the observed variation (permutation ANOVA $R^2$ = 0.51, $P$ < 0.001). (B) Meningococcal strain had no significant effect. (C) Squared cosines of the included cytokines across the different principal components.

and possibly by cell death. Additionally, the observed baseline TEER variation between donors is likely a result of varying populations of epithelial cell types and the quantity and quality of tight and adherence junctions. Recent meningococcal infection studies using ALI cultures of Calu-3 cells showed varied effects of infection on epithelial barrier function. Peters et al. observed no TEER change during infection for five meningococcal strains (22). Epithelial transcription of tight junction proteins ZO-1 and occludin, and permeability to FITC-dextran were also not affected by infection. This might be due to the use of a lower MOI of 0.1 possibly resulting in lower bacterial loads in general, or shorter incubation times at cytotoxic levels of bacteria. It could also be a result of the different *N. meningitidis* strains tested. Dave et al. infected ALI Calu-3 cultures with different carriage strains at an MOI of ~30 (23). They did not report TEERs in the context of infection but reported cytotoxicity and/or increase of FITC-dextran permeability for multiple strains, indicative of a decreased epithelial barrier function. This effect was dependent on *pilE* expression, as deletion partially restored barrier function. For MenB MC58, the only strain used in both studies, no cytotoxicity or change in permeability was reported, corroborating with the results of Peters et al. Peters et al. also reported the invasion of meningococci into the basolateral chamber and the intracellular space,

even on inserts where no damage to the epithelial barrier was observed. In contrast, Audry et al., also using ALI Calu-3 cultures, observed very little basolateral translocation, even when cell layers were permeabilized by IL-4 or IL-13 treatment (24). We did not examine intracellular bacteria and hardly observed any CFUs in the basolateral chambers of the primary epithelial ALI cultures (data not shown), although this could have also been due to the small pore size (0.4 µm) of the inserts. Taken together, these and our results suggest that epithelial barrier integrity is affected by bacterial load, but that it is also dependent on strain-specific effects. In addition, at least in ALI Calu-3 cultures, meningococci seem to translocate to the basolateral site via an intracellular route similar to what Sutherland et al. had shown previously for liquid-to-liquid interface Calu-3 cultures (25). Infection of primary nasopharyngeal and bronchial epithelial cells by *Neisseria subflava* (MOI = 10) also caused a decrease in TEER, even though it is a *Neisseria* species commonly found in the nasal microbiota that only rarely causes an opportunistic infection, although the authors show an involvement for it in bronchiectasis (26).

When examining bacterial binding 2 h after infection, we observed subtle differences between strains and donors. The MenW cc11 strain showed significantly higher binding than the MenW MP01240070 strain, but no differences between the two MenC strains were observed. Meningococcal adhesion was highest in the unencapsulated strain, suggesting that adhesion and a protective capsule compete, conforming with general consensus and previous observations of downregulation of capsule during close association with host cells (27, 28). However, because the unencapsulated strain was not an isogenic mutant of any of the other tested strains, we cannot exclude that the observed binding differences are a result of other isolate-specific factors. Epithelial cells from different donors could differ in the number of adhesin-receptors on the apical membrane and mucus production, which could possibly explain our observed differences in meningococcal binding. Bacterial growth on the epithelial layer was notably different between strains and donors and showed an increased importance for specific donor-strain interactions. MenW cc11 and cnl exhibited relatively consistent growth levels across all donors, whereas MenW MP01240070 and MenC cc11 displayed over $10^5$-fold differences in growth across different donors, suggesting that growth is influenced by intrinsic bacterial factors and, to differing degrees, on the strain-specific interactions with the epithelial layers. Pathogen-associated molecular patterns receptor and/or immune factor receptor expression and amount of AMPs in the lumen could be behind the observed donor-associated differences. Bacterial cell association assessed at 6 and 24 h, or 18 h by Dave et al. did not differ between the examined strains (23). Peters et al. reported minor but significant differences between some of the strains 24 h after infection (22). They showed a ~1,000-fold increase (~10 rounds of replication) of meningococci in 24 h, much higher than our reported 1- to 100-fold increase in 24 h in most conditions, but this might be due to their 10-fold lower inoculum dose. Dave et al. reported the number of cell-associated bacteria in CFU/mL, without providing the volume collected, making direct comparisons difficult. However, through gene deletion, the number of cell-associated bacteria was shown to be partially dependent on adhesion-associated proteins, type IV protein PilE, NadA, and Opa proteins, in decreasing order of dependence. Comparison of apically secreted proteins of Calu-3 cells and primary normal human bronchial epithelial cells (29) showed secretion of AMP by the primary cells, but not the Calu-3 cell line. Likely, our primary nasal epithelium also secreted AMPs that could limit meningococcal growth, which could explain different observations with the Calu-3 cell experiments, such as the large discrepancy between our observed meningococcal growth in 24 h and that of Peters et al.

Upon meningococcal infection, we observed a mostly universal induction of CCL20, CXCL1, CXCL8, CXCL10, and IL-18 secretion 24 h post-infection, regardless of the specific meningococcal strain used for infection. These cytokines could act, among others, as signals to recruit and activate neutrophils, T-cells, and NK-cells, and induce epithelial repair (30–34). The observed cytokine induction is possibly specific to the meningococcal infection, as we have recently shown that *S. pneumoniae* induced very little cytokine

secretion (van Woudenbergh et al. submitted [21]) in the same epithelial cell model. This also suggests that *N. meningitidis* does little to suppress the response, in contrast to *Neisseria lactamica*, which was shown to inhibit cytokine release by Detroit-562 cells (35). Interestingly, we observed a significantly higher CCL20 release upon 24 h of infection by the MenW cc11 strain, which could be indicative of increased cell stress in this condition or of a strain-specific induction. CCL20 can recruit Th17 cells, B cells, and dendritic cells (31), recruit granulocytes in pneumococcal meningitis (36), can have antimicrobial activity (37), and could induce MUC5AC mucin production in a human airway epithelial cell line (38). However, after 48 h, CCL20 levels were similar to those in wells infected with other strains. Furthermore, in our PCA analysis, we observed that the overall cytokine profile observed was largely dependent on epithelial cell donor, likely a result from the previously described expected difference in epithelial layer cell constitution and gene expression. This resulted in distinct donor cytokine secretions such as IL-6, CXCL5, and CCL2, which could contribute to host-specific immune signals. Further assessment of host-related differences would require larger experiments including more donors. Furthermore, as we have examined a preselected set of cytokines, we might have missed other responses that are of importance for preventing invasive meningococcal infection. Peters et al. reported increased levels of CCL20 and CXCL8, and increased transcription of CXCL1, CXCL2, CXCL8, IL-6, and CCL20, confirming some of our observations (22). However, Audry et al. reported increased transcription of TNFα, IL-12p70, and IFNγ, but not CXCL8 (24), suggesting that cytokine secretion is dependent on experimental setup. The remarkable differences between the studies described and compared to our contrasting findings of meningococcal binding when using cell lines in liquid culture or ALI primary epithelial cultures highlight the importance of using models that more closely mimic the *in vivo* environment and to confirm results with *in vivo* observations.

The mechanisms that have been described for invasive strains, including cc11 lineage strains, sketch an evolutionary strategy of closer interaction with the host epithelial layer compared to carriage strains. Most cc11 strains express adhesin NadA, which is only found in 5% of carriage isolates (39). We observed greater binding by MenW but not MenC cc11 compared to reference strains. In addition, cc11 strains contain multiple Meningococcal Disease Associated phage (MDAφ) copies, which when produced by the bacteria form a biofilm that is different from the biofilms formed by extracellular DNA which is associated with carriage (11), increasing colonization rate and bacterial load (40). MenC but not MenW cc11 showed greater growth than reference strains (*P* = 0.090). Living close to the host cells exposes the bacterium to more immune factors, requiring the evolution of protective mechanisms such as the polysaccharide capsule, which almost all invasive isolates express while carriage isolates are often unencapsulated. Additionally, Spoerry et al. showed that invasive meningococci were more likely, including all examined cc11 strains, to encode IgA1 proteases which can both cleave sIgA1, as well as IgG3, a potent inducer of opsonophagocytosis (14). Induction of an immune response and inflammation can also result in epithelial barrier leakage, causing the release of nutrients which allows for enhanced growth and increases the likelihood of transmission to a new host. The increased bacterial load, as well as short stints of meningococcal subepithelial and/or intracellular survival (39, 41), increases the likelihood of progression into growth within the host, leading to the various debilitating results we have named IMD.

Limitations of this study are the relatively small number of strains that have been tested, making it difficult to distinguish between individual strain effects and the effect of meningococcal factors such as serogroup and clonal complex. Additionally, we did not assess the status of phase variation of meningococcal protein expression. Differences in the expression of important adhesins could have impacted our observed binding data. Also, despite the use of primary cells, conditions *in vitro* may not reflect the *in vivo* situation. In the future, this may be assessed by examining both the tissues of prospective donors and the resulting epithelial layers in culture, although obtaining the required quantity and intactness of harvested tissue could prove difficult. Our model does not

include the multitude of immune and other cells located locally in the mucosa. This allows us to examine the signals sent by epithelial cells upon recognition/presence of the meningococcus, without the complication of interaction and feedback of the immune system. However, the interplay between the immune system and the epithelial cells might be of crucial importance during infection, potentially resulting in differences in the epithelial response and directly affecting survival. It would be interesting to examine in future experiments the possibilities of mimicking this interplay by coculturing the epithelial cells with immune cells, to further expand the model. Another important factor that is not present in our model is the other microbes that reside on and interact with the nasal epithelium. The presence of other commensal or pathogenic bacteria could affect the inflammatory state of the epithelium. More importantly, the presence of pathogenic respiratory viruses will probably play an important role in the barrier integrity of the epithelial layer, enabling invasion of the host circulation (42, 43).

To conclude, we observed that invasive cc11 isolates negatively impacted barrier function, which was associated with bacterial loads. Only slight differences in epithelial immune responses were observed, implying that similar immune recruitment following meningococcal colonization by different isolates is to be expected. This would suggest that the invasiveness of cc11 strains is mostly determined by other aspects of the infection, such as differences in adaptive immune responses further down the line, or the well-documented importance of a functioning complement system. Possibly, this increased damage to the epithelial barrier without causing an increased epithelial response is an important factor that makes this lineage hyperinvasive. Interestingly, in our study, it was the specific combinations of epithelial cell donors and meningococcal strains that determined barrier integrity and bacterial growth during infection. Possibly, such specific effects also determine which meningococcal colonization events result in IMD. For example, MenC cc11 showed a large variation in bacterial load, depending on the epithelial cell donor. Perhaps it is only invasive when it resides on an epithelium that supports vigorous growth. Therefore, further studies investigating how this interaction may explain the risk of IMD are warranted, which might result in new or more inter-action-specific treatment and/or prevention strategies and thereby a decrease in IMD burden.

## MATERIALS AND METHODS

### Isolation of primary nasal epithelial cells

Post-operative residual tissue was collected from seven individuals at the Radboudumc, an academic hospital in the Netherlands. Voluntary informed consent was provided prior to the surgery. No macroscopical abnormalities could be observed in the residual tissue. Obtained tissues were washed with PBS supplemented with 200 IU/mL penicillin/streptomycin (Cambrex-Sigma) and 0.2 mg/mL gentamycin (Sigma). When required, tissues were cut into smaller pieces prior to incubation in Hank's balanced salt solution (Gibco) supplemented with 0.02 mg/mL DNAse1 (Sigma-Aldrich) and 1 mg/mL collagenase (Sigma) for 1.5 h at 37°C while rotating. The resulting cell suspension was filtered through a 70 µm strainer, leaving only single cell-sized or smaller particles. Cells were then centrifuged at $400 \times g$ for 15 min at 10°C and resuspended in 10 mL of expansion medium (PneumaCult-Ex Basal medium; Stemcell Technologies) supplemented with hydrocortisone (96 ng/mL; Stemcell), penicillin/streptomycin (100 IU/mL), and gentamycin (0.1 mg/mL). Cultures were incubated at 37°C in 5% $CO_2$ until cells reached confluency. At passage 2, cells were suspended at a density of $1 \times 10^6$ cells/mL in Ham's nutrient mixture F12 (Merck) supplemented with 10% fetal calf serum (HyClone), 10% dimethyl sulfoxide (Sigma-Aldrich), 2% 1.5 M HEPES (Sigma-Aldrich), 100 IU/mL penicillin/streptomycin and 0.1 mg/mL gentamycin, and stored at −135°C until further use.

## Epithelial cell culture

Frozen stocks were used to inoculate T-75 flasks containing PneumaCult-Ex Plus Medium (StemCell) supplemented with PneumaCult-Ex Plus Supplement 50× (Stem-Cell), Hydrocortisone Stock Solution 200× (StemCell) and penicillin/streptomycin 100× (Sigma-Aldrich). Cells were incubated at 37°C in 5% $CO_2$, and medium was replaced every other day until the cells were 80–90% confluent. To prepare for ALI-cell cultures, inserts (0.4 µm pores) of transparent 24-well plates (Corning) were coated with 100 µL collagen type I (rat, Gibco; 0.03 mg/mL collagen) and incubated at 37°C for 45 min, followed by washing with PBS. Then, growth medium of confluent cell cultures was removed and, after washing with HEPES-BSS (Lonza cc-5024), replaced by trypsin EDTA (0.025%, Lonza CC-5012). After cell detachment, trypsin neutralizing solution (TNS; Lonza CC-5002) was added, and the cell suspension was collected, centrifuged ($450 \times g$), and resuspended in supplemented PneumaCult-Ex Plus Medium. Cells were counted using trypan blue, and 100 µL of $1 \times 10^6$ cells/mL was added to the apical side of the coated inserts. After the addition of 500 µL of supplemented PneumaCult-Ex Plus Medium to the basolateral side, plates were incubated at 37°C in 5% $CO_2$. Once cells were attached, apical and basolateral cell medium was replaced every other day until a tight monolayer had formed. Then, apical and basolateral medium was removed, and differentiation medium was added to the basolateral side only: PneumaCult-ALI basal medium supplemented with PneumaCult-ALI supplement 10×, PneumaCult-ALI maintenance supplement 100× (StemCell), Hydrocortisone Stock Solution 200× (StemCell), Penicillin-Streptomycin 100× (P/S, Sigma-Aldrich) and heparin (0.2%) 500× (StemCell). The differentiation medium was replaced three times a week. After 2–3 weeks, the epithelial layers produced significant amounts of mucus, which was washed away weekly, by apical administration and removal of 100 µL PBS. Epithelial cell layers were differentiated at the ALI for 4–6 weeks. Following differentiation, the number of cells per insert ranged from ~$1 \times 10^5$ to ~$3 \times 10^6$, depending on the epithelial cell donor.

## *N. meningitidis* culturing and infection of epithelial cell cultures

Table 2 describes the *N. meningitidis* isolates used for the main (strains 1–5) and supplemental (strains 6–11) figures of this study. Meningococci were grown O/N at 37°C, 5% $CO_2$ on Columbia blood agar plates (7% sheep blood), and afterward collected and diluted in PBS to the required concentration based on $OD_{600}$. Prior to infection, epithelial cells were cultured in differentiation medium without P/S and heparin for 1 day. On the day of infection, the basolateral medium was replaced with 1 mL differentiation medium without P/S and heparin. Afterward, epithelial cell layers were washed apically with 100 µL PBS to remove mucus, and cells were infected with ~$1 \times 10^5$ CFU in 25 µL of differentiation medium. Uninfected wells received 25 µL of differentiation medium only apically. Data from seven donors was obtained in two independent experiments. D1–D4 were infected with strains 1–5 and D5–D7 with strains 1–11.

To determine the number of bacteria bound at designated time points, cell layers were washed and treated with 100 µL PBS with 1% saponin for 10 min to lyse the epithelial cells. Epithelial layers were then scraped, and the resulting suspension was titrated and plated on blood agar plates O/N at 37°C, 5% $CO_2$. Meningococcal colonies were then counted. All CFU measurements were performed singly.

For RPMI-2560 and Calu-3 binding experiments, cells were cultured in DMEM supplemented with 10% FCS and 1% NEAA. In 96-well plates, per well, $2 \times 10^5$ cells were infected with an MOI of 0.1, 1 or, 10 in 100 µL of DMEM. Binding was assessed as described above.

## TEER measurement

TEER measurements were performed using a Millicell ERS-2 Voltohmmeter (Merck). At designated time points, 100 µL of PBS was added to the apical side of epithelial layers, and the electrode was used to determine the resistance (Ω) across the cell layer.

The electrical resistance of inserts without cells was subtracted from measured values. Resistance is reported as $\Omega \times$ area (cm$^2$), with an insert area of 0.33 cm$^2$. TEER at $t = 0$ was measured before addition of inoculum, resulting in $n = 24$ (D1–D4) and $n = 36$ (D5–D7) replicates. At $t = 24$, TEER was measured in triplicate for D1–D4 or singly for D5–D7.

## Cytokine measurement

A total of 100 µL medium from the basolateral side of the epithelial ALI layer was collected 6 h (D1–D4: $n = 3$; D5–D7: $n = 2$), 24 h (D1–D4: $n = 3$; D5–D7: $n = 2$), or 48 h (D1–D7: $n = 1$) after infection. Cytokine concentrations in 10× dilutions of the collected samples were measured using a BD FACS Canto II flow cytometer and two LEGENDplex kits (Human Proinflammatory Chemokine Panel 1-740985, Human Inflammation Panel 1-740809), which measure the following 24 cytokines: CXCL1, CXCL5, CXCL8, CXCL9, CXCL10, CXCL11, CCL2, CCL3, CCL4, CCL5, CCL17, CCL20, IL-1β, IL-6, IL-10, IL-12p70, IL17A, IL-18, IL-23, IL-33, IFNα2, IFNγ, and TNFα according to the manufacturer's protocol.

## Data analysis

Microsoft Excel, GraphPad Prism, R, and R-studio were used for data storage, visualization, script writing, and statistical analyses. The mean baseline TEER of donor epithelial layers was compared using a Kruskal-Wallis test. For mean TEER comparison after infection, Friedman permutation tests were performed ($1 \times 10^5$ permutations), where data were stratified by either epithelial cell donor or meningococcal strain. These were followed by post hoc Dunn's tests with Benjamini-Hochberg multiple testing correction. Percentage of inoculum bound and CFU fold change were determined for each donor-strain combination and compared using Friedman permutation tests using the same method. To assess whether TEER levels were dependent on CFU load, log-transformed linear regression was performed.

Cytokine concentrations were calculated using standard curves of stock cytokine dilutions. Measurements that were at the edges of the standard curve, which could not be reliably assigned a value, were assigned the lowest or highest measured value of that cytokine, respectively. Mean cytokine log$_{10}$-transformed concentration was calculated for each specific donor-strain combination. Friedman permutation tests were performed for the $t = 6$ and $t = 24$ time points of each cytokine, with Benjamini-Hochberg multiple testing correction. When significant differences were observed, a post hoc Dunn's test with Benjamini-Hochberg multiple testing correction was performed. PCA was performed on the 24 h post-infection concentrations of a selection of cytokines that could be detected in all conditions. PERMANOVA was performed (9,999 permutations) to quantify how much of the variation could be explained by epithelial cell donor or meningococcal strain. All data are available under Supplemental material.

## ACKNOWLEDGMENTS

Mirjam Knol for providing clinical information for strain selection, Willem Miellet for providing the cnl strain, NLRBM for providing the invasive isolates, and Mioara Alina Nicolae for advice on statistical testing.

## AUTHOR AFFILIATIONS

[1]Center for Immunology of Infectious Diseases and Vaccines, National Institute for Public Health and the Environment, RIVM, Bilthoven, the Netherlands
[2]Laboratory of Medical Immunology, Radboud UMC, Nijmegen, the Netherlands

## PRESENT ADDRESS

Milou Ohm, Amsterdam University Medical Center, Amsterdam, the Netherlands

## AUTHOR ORCIDs

Daan W. Arends ⓘ http://orcid.org/0000-0002-2719-6408
Marien I. de Jonge ⓘ http://orcid.org/0000-0003-2812-5895
Gerco den Hartog ⓘ http://orcid.org/0000-0002-2103-6315

## FUNDING

| Funder | Grant(s) | Author(s) |
| --- | --- | --- |
| Horizon 2020 Framework Programme | 835433 | Gerco den Hartog |
| Ministry of Health, Welfare and Sports | NA | Daan W. Arends |
| | | Debbie van Rooijen |
| | | Esther van Woudenbergh |
| | | Janine Wolf |
| | | Milou Ohm |
| | | Gerco den Hartog |

## AUTHOR CONTRIBUTIONS

Daan W. Arends, Formal analysis, Visualization, Writing – original draft, Writing – review and editing | Debbie van Rooijen, Data curation, Investigation, Methodology, Writing – review and editing | Esther van Woudenbergh, Conceptualization, Investigation, Methodology, Resources, Writing – review and editing | Janine Wolf, Investigation, Methodology | Milou Ohm, Investigation, Methodology, Writing – review and editing | Marien I. de Jonge, Resources, Supervision, Writing – review and editing | Gerco den Hartog, Conceptualization, Data curation, Formal analysis, Funding acquisition, Methodology, Project administration, Resources, Supervision, Writing – review and editing

## ADDITIONAL FILES

The following material is available online.

### Supplemental Material

**Supplemental figures (Spectrum00141-25-s0001.docx).** Fig. S1 to S3.
**Supplemental material (Spectrum00141-25-s0002.xlsx).** Experimental data

### Open Peer Review

**PEER REVIEW HISTORY (review-history.pdf).** An accounting of the reviewer comments and feedback.

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
