## [Reviewer comments · Microbiology Spectrum]

Microbiology Spectrum

Impact of Host Factors and Invasive Meningococci on Bacterial Adhesion, Proliferation, Primary Nasal Epithelial Barrier Function, and Immune Response

Daan Arends, Debbie van Rooijen, Esther van Woudenberg, Janine Veldman-Wolf, Milou Ohm, Marien de Jonge, and Gerco den Hartog

Corresponding Author(s): Daan Arends, Rijksinstituut voor Volksgezondheid en Milieu Centrum Infectieziektebestrijding

Review Timeline:

Submission Date:	January 15, 2025
Editorial Decision:	March 10, 2025
Revision Received:	May 12, 2025
Accepted:	May 27, 2025

Editor: Alex Dulovic

Reviewer(s): Disclosure of reviewer identity is with reference to reviewer comments included in decision letter(s). The following individuals involved in review of your submission have agreed to reveal their identity: Mathieu Coureuil (Reviewer #1); Christopher D Bayliss (Reviewer #2)

Transaction Report:

DOI: <https://doi.org/10.1128/spectrum.00141-25>

Re: Spectrum00141-25 (Impact of Host Factors and Invasive Meningococci on Bacterial Adhesion, Proliferation, Primary Nasal Epithelial Barrier Function, and Immune Response)

Dear Mr. Daan Wouter Arends:

Thank you for the privilege of reviewing your work. Below you will find my comments, instructions from the Spectrum editorial office, and the reviewer comments.

Based upon the reviewer reports as well as my own assessment of the manuscript, the decision is MODIFICATIONS. You will find the reviewers comments at the bottom of the email. There are no serious concerns with the work, but both reviewers have listed several points that could be evaluated as part of the discussion.

Revision Guidelines

Sincerely,
Alex Dulovic
Editor
Microbiology Spectrum

Reviewer #1 (Comments for the Author):

This works focuses on different serogroup W and C meningococcal strains of cc11 or cc22 and a non-capsulated isolate. The study attempts to determine whether cc11 isolates would have different phenotypes in terms of impact on TEER of an ALI

primary cell culture model, as well as cytokine secretion or colonization.

Overall, this work has no major shortcomings apart from the absence of striking results, but this is also the risk of an exploratory science. The authors have chosen to publish these results, which is a contribution to the scientific community.

I have no major comments to make, but three that deserve to be discussed by the authors.

- There are contradictory evidences on results obtained with ALI models and Nm by the different teams (TEER, Growth ...). This may be due to the production of antimicrobial peptide, especially by primary cells, this should be discussed. One limitation is that the role of the microbiota is not assessed here, and this may be discussed.

- It is not clear if the ALI culture cells produce mucus or not? This point should be discussed with regard to the "binding" experiment. Do bacteria bind to cells or are they trapped into the mucus?

- The use of the cni isolate is interesting but the role of the capsule using a non-capsulated isolate instead of a non-capsulated mutant derivative prevents a detailed analysis. This limitation should be discussed.

Reviewer #2 (Comments for the Author):

This article by Arends et al. tackles the important but difficult issue of the responses of epithelial cells to meningococcal infection. The authors have utilised primary nasal cells for this study, providing an approximation of events that are actually data occurring during infections in humans and improves on experiments with cell lines. The experiments are clearly described and appropriate analyses are performed. The interpretation of the findings is informed and draws some useful points (which I have summarised below to make sure I understood what the authors were saying) and comparisons to published literature. The limitations of the study are also clearly described with perhaps more emphasis required on the variable composition of the different cultures and a missing comment on phase variation in meningococcal adhesins.

The key results appear to be that 1) basal TEER measurements for nasopharyngeal cells differ between donors, 2) all meningococcal strains, including a capsule null strain, produce similar reductions in TEER on most donor cell lines, 3) cell lines respond in different ways to infection (differences in TEER values, bacterial adherence/growth, cytokine production), 4) capsule null strains have higher adherence than other meningococci, 5) there is a weak effect of load on TEER measurements, 7) donor line has more effect on outcome than strain.

Major

Lines 98 and 190. It should be emphasised in both of these primary lines are mixtures of different types of nasal epithelial cells (and possibly of other cell types?). May be also indicated as being from 'healthy' tissue if that is the case. As these tissues are relevant to meningococcal host colonisation, it is important to emphasize this point. Was any staining of the tissues performed to assess the tissue types or organisation after formation of the barrier? If so this could be included to provide an indication of the nature of these cell barriers.

Line 298. The limitations should include the lack of data on the phase variation expression states of relevant adhesins, particularly the Opa and Nad proteins. Differences in the expression states may have impacted adhesion levels for each strain.

Line 330. The specific tissue used as the source for the post-operative residual tissue needs to be clarified. Presumably this was nasal tissue as the authors indicate that they were working on nasal epithelial primary cells but the source should be stated.

Minor

Line 46. "gram-negative" should be "Gram-negative"

Line 49. What does the "it" refer to?

Line 61/62. The statement of expression of capsules is poorly expressed as it implies that single isolates can produce more than one capsular antigen. This sentence should indicate that cc11 have undergone frequent capsule switching so that these isolates have been associated with the following serogroups

Line 67. Type IV pili would be better than pilus proteins.

Lines 152-157. Fig 3B provides an excellent overview of all the data for adhesion, load and TEER measurements. The authors provide some salient summary points that seem justified. One consideration is whether there is a universal effect and looking at the data it looks like cell lines D4 and D6 respond to any infection with similar decreases in TEER measurements that are greater than occurs in the uninfected. While this point is already captured in Fig 1D, it may be worth re-iterating here.

Line 163-164. May be useful to indicate whether any responses were detected for major cytokines particularly TNF-alpha and IL-1-beta (or is it known that these cytokines are not produced by these types of epithelial cells).

Line 213. Dave et al did not examine invasion of bacteria into the basolateral chamber. So "Both" should be changes to "Peters et al"

Line 285. Word missing? Should "extracellular" be "extracellular DNA"?

Line 300 Reword "despite primary cells were used" to "despite the use of primary cells"

Line 308. Deleted "in" for phrase "directly in affecting survival"

Line 323. Alter to "resides on AN epithelium"

Figure 1. Panel A has multiple measurements for each donor. The total number of independent measurements needs to be indicated (presumably 12 if each infection was repeated 2 times as indicated in lines 385-386).

Figure 1. Panel B. The dashed lines are the mean of two measurements. I did wonder whether all 12 measurements should be shown individually but this would probably clutter up the figure. I assume the raw data will be provided as a linked resource.

Figure 1. Legend. The p values for the asterisks need to be indicated in the legend. It should also be stated that lines are grouped by p value, which is the reason asterisks are not present on all comparison lines.

Figure 2. The amount in CFU and MOI of the inoculum should be indicated in the legend.

Response to reviewers

Dear Alex Dulovic and Reviewers,

We sincerely thank you for providing us with the opportunity to improve our manuscript, titled "Impact of Host Factors and Invasive Meningococci on Bacterial Adhesion, Proliferation, Primary Nasal Epithelial Barrier Function, and Immune Response" (Spectrum00141-25R1). We appreciate the time and effort taken to evaluate our work.

In this revised version, we have addressed each comment raised by the reviewers and have made the corresponding changes to the manuscript. Below, we provide a detailed response to the reviewers' comments.

➤ Reviewer #1 (Comments for the Author):

This work focuses on different serogroup W and C meningococcal strains of cc11 or cc22 and a non-capsulated isolate. The study attempts to determine whether cc11 isolates would have different phenotypes in terms of impact on TEER of an ALI primary cell culture model, as well as cytokine secretion or colonization.

Overall, this work has no major shortcomings apart from the absence of striking results, but this is also the risk of an exploratory science. The authors have chosen to publish these results, which is a contribution to the scientific community.

I have no major comments to make, but three that deserve to be discussed by the authors.
- There are contradictory evidences on results obtained with ALI models and Nm by the different teams (TEER, Growth ...). This may be due to the production of antimicrobial peptide, especially by primary cells, this should be discussed. One limitation is that the role of the microbiota is not assessed here, and this may be discussed.

Thank you for your comment, we agree that the roles of AMP and microbiota are important in *in vivo* colonization and infection processes. Therefore, we have added a section about AMPs in the discussion, including an additional reference (lines 262-266):

Comparison of apically secreted proteins of Calu-3 cells and primary normal human bronchial epithelial (NHBE) cells [30] showed secretion of AMP by the primary cells, but not the Calu-3 cell line. Likely, our primary nasal epithelium also secreted AMPs that could limit meningococcal growth, which could explain different observations with the Calu-3 cell experiments, such as the large discrepancy between our observed meningococcal growth in 24h and that of Peters *et al.*.

30. Sanchez-Guzman, D., et al., *Long-term evolution of the epithelial cell secretome in preclinical 3D models of the human bronchial epithelium*. *Sci Rep*, 2021. **11**(1): p. 6621.

We also added a section about the limitation of the absence of the other microbiota in the nose, also including additional references (lines 328-333):

Another important factor that is not present in our model is the other microbes that reside on and interact with the nasal epithelium. The presence of other commensal or pathogenic bacteria could affect the inflammatory state of the epithelium. More importantly, the presence of pathogenic respiratory viruses will probably play a role

in the barrier integrity of the epithelial layer, enabling invasion of the host circulation [43, 44].

43. Li, X., et al., *The intricate interplay among microbiota, mucosal immunity, and viral infection in the respiratory tract*. Journal of Translational Medicine, 2025. **23**(1): p. 488.

44. Jansen, A.G.S.C., et al., *Invasive pneumococcal and meningococcal disease: association with influenza virus and respiratory syncytial virus activity?* Epidemiology and Infection, 2008. **136**(11): p. 1448-1454.

- It is not clear if the ALI culture cells produce mucus or not ? This point should be discussed with regard to the "binding" experiment. Do bacteria bind to cells or are they trapped into the mucus?

We indeed did not explicitly mention mucus production. The epithelial layers start to produce notable amounts of mucus around week 2-3 of differentiation. Because the mucociliary clearance in vitro does not lead to removal of the mucus due to the constraints of the transwell insert, from then on, epithelial cells are washed weekly with 100 μ L PBS to prevent excess mucus build up. Prior to inoculation, all inserts were also washed with 100 μ L PBS. This last wash was described in the materials and methods, without mention of mucus. We have added this (lines 409-411), as well as the weekly wash to remove excess mucus (lines 390-391). Because this mucus was removed prior to performing the experiment, we assume the observed bound CFUs to be adhering to the epithelial cells and not being entrapped within freshly secreted mucus.

- The use of the cnl isolate is interesting but the role of the capsule using a non-capsulated isolate instead of a non-capsulated mutant derivative prevents a detailed analysis. This limitation should be discussed.

Agreed. We have added a sentence describing the limitation (lines 241-243).

➤ Reviewer #2 (Comments for the Author):

This article by Arends et al. tackles the important but difficult issue of the responses of epithelial cells to meningococcal infection. The authors have utilised primary nasal cells for this study, providing an approximation of events that are actually data occurring during infections in humans and improves on experiments with cell lines. The experiments are clearly described and appropriate analyses are performed. The interpretation of the findings is informed and draws some useful points (which I have summarised below to make sure I understood what the authors were saying) and comparisons to published literature. The limitations of the study are also clearly described with perhaps more emphasis required on the variable composition of the different cultures and a missing comment on phase variation in meningococcal adhesins.

The key results appear to be that 1) basal TEER measurements for nasopharyngeal cells differ between donors, 2) all meningococcal strains, including a capsule null strain, produce similar reductions in TEER on most donor cell lines, 3) cell lines respond in different ways to infection (differences in TEER values, bacterial adherence/growth, cytokine production), 4) capsule null strains have higher adherence than other meningococci, 5) there is a weak effect of load on TEER measurements, 7) donor line has more effect on outcome than strain.

Major

Lines 98 and 190. It should be emphasised in both of these primary lines are mixtures of different types of nasal epithelial cells (and possibly of other cell types?). May be also indicated as being from 'healthy' tissue if that is the case. As these tissues are relevant to meningococcal host colonisation, it is important to emphasize this point. Was any staining of the tissues performed to assess the tissue types or organisation after formation of the barrier? If so this could be included to provide an indication of the nature of these cell barriers.

We have examined the epithelial layers by (fluorescence) microscopy, staining for tubulin IV (cilia), MUC5AC (goblet cells) and SCGB1A1 (club cells). Here we observed that 6 week old epithelium from donors D3 and D5 had little staining for cilia, whereas all others showed abundant ciliated cells. For the other cell types less difference was observed. D1, D2 and D7 showed slightly more secretory SCGB1A1-positive cells. And D3 and D5 showed slightly fewer goblet (MUC5AC) cells. Examination of cross sections showed that epithelial layers had similar thickness, with the exception of D5 which was much thinner. We have not included these images, as colleagues of mine have submitted a manuscript containing these images:

“Primary human nasal, nasopharyngeal, and bronchial epithelia show distinct immune responses to various pathogens”

21. Van Woudenberg, E., et al., *Primary human nasal, nasopharyngeal, and bronchial epithelia show distinct immune responses to various pathogens*. Pathogens and Disease, submitted.

In it, primary epithelial layers of nasal, nasopharyngeal and bronchial origin are compared. To not publish the same data twice, we excluded it, but to provide some of this information to the readers we added a brief description to the beginning of the results section (lines 99-103).

Line 298. The limitations should include the lack of data on the phase variation expression states of relevant adhesins, particularly the Opa and Nad proteins. Differences in the expression states may have impacted adhesion levels for each strain.

We agree, and have added the limitation to our discussion (line 316-318).

Line 330. The specific tissue used as the source for the post-operative residual tissue needs to be clarified. Presumably this was nasal tissue as the authors indicate that they were working on nasal epithelial primary cells but the source should be stated.

We have added an additional column in table 1, giving more details on the tissue source within the nasal cavity.

Minor

Line 46. "gram-negative" should be "Gram-negative"

Corrected the error (line 47).

Line 49. What does the "it" refer to?

“It” refers to colonization, with which we have replaced the word (line 49).

Line 61/62. The statement of expression of capsules is poorly expressed as it implies that single isolates can produce more than one capsular antigen. This sentence should indicate that cc11 have undergone frequent capsule switching so that these isolates have been associated with the following serogroups

We agree that the formulation was poor, and have improved the sentence to reduce the chance of misinterpretation (lines 62-64).

Line 67. Type IV pili would be better than pilus proteins.

We changed it to type IV pili (line 68).

Lines 152-157. Fig 3B provides an excellent overview of all the data for adhesion, load and TEER measurements. The authors provide some salient summary points that seem justified. One consideration is whether there is a universal effect and looking at the data it looks like cell lines D4 and D6 respond to any infection with similar decreases in TEER measurements that are greater than occurs in the uninfected. While this point is already captured in Fig 1D, it may be worth re-iterating here.

We appreciate your observation, but have decided against re-iteration. Indeed, there are similar effects of infection on TEER for donors D4 and D6. However, as only 2 out of 7 donors show this similarity, we have decided not to highlight this specific universal response.

Line 163-164. May be useful to indicate whether any responses were detected for major cytokines particularly TNF-alpha and IL-1-beta (or is it known that these cytokines are not produced by these types of epithelial cells).

We have added a part on TNF α and IL-1 β , and added IL-6 as well. The response upon infection we observed for these cytokines was not there, or not as clear as for the other cytokines. However, we agree that the importance of these cytokines in inflammation (specifically, in this case during bacterial infection) and the fact that we included the cytokines in our measurements, made mentioning these cytokines an improvement of our results section (line 177-181).

Line 213. Dave et al did not examine invasion of bacteria into the basolateral chamber. So "Both" should be changes to "Peters et al"

Thank you for showing our mistaken referral to Dave *et al*. We have corrected our error. (line 222).

Line 285. Word missing? Should "extracellular" be "extracellular DNA"?

We have added the correctly identified missing "DNA" (line 301).

Line 300 Reword "despite primary cells were used" to "despite the use of primary cells"

Line 308. Deleted "in" for phrase "directly in affecting survival"

Line 323. Alter to "resides on AN epithelium"

We made the recommended alterations (line 318, 326 and 346).

Figure 1. Panel A has multiple measurements for each donor. The total number of independent measurements needs to be indicated (presumably 12 if each infection was repeated 2 times as indicated in lines 385-386).

Figure 1. Panel B. The dashed lines are the mean of two measurements. I did wonder whether all 12 measurements should be shown individually but this would probably clutter up the figure. I assume the raw data will be provided as a linked resource.

Figure 1. Legend. The p values for the asterisks need to be indicated in the legend. It should also be stated that lines are grouped by p value, which is the reason asterisks are not present on all comparison lines.

Figure 2. The amount in CFU and MOI of the inoculum should be indicated in the legend.

We sincerely thank you for your comments on the figures. It showed that we needed to give more details on the experimental setup. We have indicated for each type of measurement the number of measurements made in the materials and methods section (TEER: line 430-432; CFU: 417-418; Cytokines: 435-436). To further explain fig. 1A, we measured the $t = 0$ TEER in quadruplicate for D1-D4 and triplicate for D5-D7. We had 6

conditions for D1-D4 (uninfected, and infection with five strains) and 12 conditions for D5-D7 (uninfected, infection with the same five strains, and infection with 6 additional strains). As $t = 0$ TEER was measured before conditional treatments were applied, we had six quadruplicates for D1-D4 and twelve triplicates for D5-D7, our resulting total measurements per donor were 24 for D1-D4 and 36 for D5-D7.

For our CFU measurements, epithelial layers had to be sacrificed, resulting in a loss of inserts, where quadruple replicates turn in to triplicates after the 2h binding measurement, and to single measurements after 24h CFU measurement and fixation for microscopy (the latter yielded no interesting results, so was left out of the manuscript). We hope the additions to the materials and methods section provides a clear picture of the number of measurements.

We included better descriptions of shown results of statical analysis for figure 1.

We have included MOI and CFU in fig. 2 legend. We also wish to apologize for an incorrect statement in the legend, referring to a mean of each donor/strain, when in fact it showed single measurements.

We hope that you find the revised manuscript to be improved and suitable for publication in Microbiology Spectrum. Thank you again for your time and consideration. We are happy to address any further questions or concerns.

Re: Spectrum00141-25R1 (Impact of Host Factors and Invasive Meningococci on Bacterial Adhesion, Proliferation, Primary Nasal Epithelial Barrier Function, and Immune Response)

Dear Mr. Daan Wouter Arends:

Thank you for submitting your revised manuscript. I am pleased to inform you that it has now been accepted for publication.

I am forwarding it to the ASM production staff. Your paper will first be checked to make sure all elements meet the technical requirements. ASM staff will contact you if anything needs to be revised before copyediting and production can begin. Otherwise, you will be notified when your proofs are ready to be viewed.

Sincerely,
Alex Dulovic
Editor
Microbiology Spectrum

Reviewer #1 (Comments for the Author):

The authors have addressed all the concerns raised by the reviewers

Reviewer #2 (Comments for the Author):

The alterations by the authors have fully addressed my comments and concerns with regard to data presentation and interpretation..